# SHARDNET: ONE FILTER SET TO RULE THEM ALL

## ABSTRACT

Deep CNNs have achieved state-of-the-art performance on numerous machine learning and computer vision tasks in recent years, but as they have become increasingly deep, the number of parameters they use has also increased, making them hard to deploy in memory-constrained environments and difficult to interpret. Machine learning theory implies that such networks are highly over-parameterised and that it should be possible to reduce their size without sacrificing accuracy, and indeed many recent studies have begun to highlight specific redundancies that can be exploited to achieve this. In this paper, we take a further step in this direction by proposing a filter-sharing approach that reformulates deep, complex CNNs as an iterative application of shallower modules (a single convolutional mapping in the simplest case). We show, via experiments on CIFAR-10, CIFAR-100, Tiny ImageNet and ImageNet that this allows us to reduce the parameter counts of networks based on common designs such as VGGNet and ResNet by a factor proportional to their depth, whilst leaving their accuracy largely unaffected. At a broader level, our approach represents a way of rethinking neural network architectures so as to leverage the scale-space regularities found in visual signals, resulting in models that are both parsimonious and easier to interpret.

## 1 INTRODUCTION

Deep CNNs have achieved state-of-the-art results on a wide range of tasks, from image understanding (Redmon & Farhadi, 2017; Jetley et al., 2017; Kim et al., 2018; Oktay et al., 2018) to natural language processing (Oord et al., 2016; Massiceti et al., 2018). However, these network architectures are often highly overparameterised (Zhang et al., 2016), and thus require the supervision of a large number of input-output mappings and significant training time to adapt their parameters to any given task. Recent studies have discovered several different redundancies in these network architectures (Garipov et al., 2016; Hubara* et al., 2018; Wu et al., 2018; Frankle & Carbin, 2019; Yang et al., 2019a;b) and certain simplicities (Pérez et al., 2018; Jetley et al., 2018) in the functions that they implement. For instance, Frankle & Carbin (2019) showed that a large classification network can be distilled down to a small sub-network that, owing to its lucky initialisation, is trainable in isolation without compromising the original classification accuracy. Jetley et al. (2018) observed that deep classification networks learn simplistic non-linearities for class identification, a fact that might well underlie their adversarial vulnerability, whilst challenging the need for complex architectures. Attempts at knowledge distillation have regularly demonstrated that it is possible to train small student architectures to mimic larger teacher networks by using ancillary information extracted from the latter, such as their attention patterns (Zagoruyko & Komodakis, 2017), predicted soft-target distributions (Hinton et al., 2014) or other kinds of meta-data (Lopes et al., 2017). These works and others continue to expose the high level of parameter redundancy in deep CNNs, and comprise a foundational body of work towards studying and simplifying networks for safe and practical use.

Our paper experiments with yet another scheme for simplifying CNNs, in the hope that it will not only shrink the effective footprint of these networks, but also open up new pathways for network understanding and redesign. In particular, we propose the use of a common set of convolutional filters at different levels of a convolutional hierarchy to achieve class disentanglement. Mathematically, we formulate a classification CNN as an iterative function in which a *small set of learned convolutional mappings* are applied repeatedly as different layers of a CNN pipeline (see Figure 1). In doing so, we are able to reduce the parameter count of the network by a factor proportional to its depth, whilst leaving its accuracy largely unaffected. We also investigate the introduction of non-shared linear

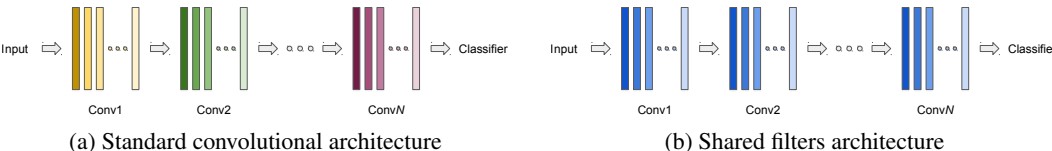

(a) Standard convolutional architecture      (b) Shared filters architecture

Figure 1: Standard CNN architectures (a) contain several convolutional layers, all of which are individually adapted using backpropagation. By contrast, we propose the use of a *single* learned convolutional layer (b) that is applied repeatedly to simulate a CNN pipeline.

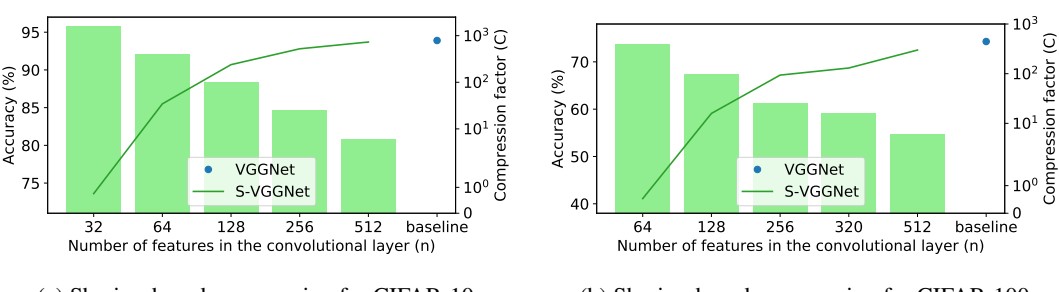

(a) Sharing-based compression for CIFAR-10      (b) Sharing-based compression for CIFAR-100

Figure 2: Accuracy versus compression trade-off curves for our basic shared architecture in Fig. 1, for different widths $n$ of the shared convolutional layer, compared to the baseline VGGNet (Simonyan & Zisserman, 2015), for CIFAR-10 (a) and CIFAR-100 (b). The compression factor is plotted on a logarithmic scale.

layers before certain shared convolutional layers to enhance the flexibility of the model by allowing it to linearly combine shared filter maps for the disentanglement task.

## 2 RELATED WORK

This work is partly inspired by the classic literature on image processing that has long sought to characterise natural images by collating their responses, at different image scales, to a small, canonical set of hand-crafted visual operators (Mallat, 1989; Viola & Jones, 2004). Modern CNN architectures (Krizhevsky et al., 2012; Simonyan & Zisserman, 2015) effectively still implement hierarchical feature extraction, but with the difference that there are thousands of such operators (i.e. convolutional filters) at each scale level, all of which are individually adaptable and learned via backpropagation. Our work can thus be seen as an effort to reconcile the above two non-contemporaneous approaches to image processing, in which we aim to identify a common set of visual operators for all the different scales by learning them in an end-to-end manner.

Our approach bears some high-level resemblance to previous approaches (e.g. Pinheiro & Collobert (2014); Liang & Hu (2015); Liao & Poggio (2016); Savarese & Maire (2019)) that have attempted to implement, interpret and potentially improve convolutional neural networks through an iterative use of simpler modules For example, Liao & Poggio (2016) share convolutional mappings in ResNets in an attempt to approximate biological visual systems using feedback loops and recurrence, although their experimental analysis is limited to the CIFAR dataset. By contrast, our work applies the convolution-sharing paradigm to both plain feed-forward and residual constructs, and investigates the effectiveness of using only a single shared convolutional mapping for the entire network pipeline. An additional contribution of our approach is the flexibility we add to the model by coupling learned linear layers with shared convolutions while still limiting the total parameter count. Experimentally, we evaluate the accuracy vs. model size tradeoff induced by our approach on a realistic set of datasets that include Tiny ImageNet and ImageNet.

A steady increase in the size of datasets and the availability of computational resources has enabled neural networks to grow deeper (Simonyan & Zisserman, 2015; He et al., 2016), denser (Huang et al., 2017) and wider (Zagoruyko & Komodakis, 2016). In doing so, concerns regarding their over-parameterisation have often been ignored in favour of better test set generalisation.[1] More

---

[1]"Despite previous arguments that depth gives regularization effects and width causes network to overfit, we successfully train networks with 5 times more parameters than ResNet-1001, . . . and outperform ResNet-1001 by a significant margin." (Excerpt from Zagoruyko & Komodakis (2016).)

recently, as their performance on some benchmarks (He et al., 2016; Oord et al., 2016) has reached near-human levels, real-world deployment of these models is being considered. This deployment has been hugely impeded by the memory requirements, latency and energy demands of their heavy computational machinery (Bianco et al., 2018).

Our approach contributes to the (extensive) literature on network compression that is focused on making these machine learning models more usable in practical scenarios. Existing compression methods can be divided into seven categories – pruning, quantisation, tensorization/tensor decomposition, knowledge distillation, custom architectures, sharing-based and hybrid methods. Many of these works are beyond the scope of this paper, but for completeness, we present a brief review in §A.1 (a more exhaustive survey can be found in Cheng et al. (2017)). Our own work falls within the realm of sharing-based methods that seek to equate some of a network's weights or filters to reduce the number of independent parameters in the network. There are various ways of deciding which weights/filters to share, from somewhat arbitrary (if effective) approaches such as the hashing trick (Chen et al., 2015; Liu et al., 2018), to more principled approaches such as k-means clustering (Wu et al., 2018). A few recent works have turned their attention to sharing convolutional weight matrices in a more structured manner. Of these, LegoNet (Yang et al., 2019b) shares filter groups across sets of channels, whilst FSNet (Yang et al., 2019a) shares filter weights across spatial locations. In both cases, sharing is restricted to a single layer at a time. ShaResNet (Boulch, 2018) reuses convolutional mappings, but within the same scale level (i.e. between two max-pooling steps). The novelty of our work lies in extending this filter-sharing paradigm to an entire convolutional pipeline. We instantiate a single convolutional layer that is applied iteratively to mimic a deep convolutional feature extractor, and analyse the accuracy vs. memory tradeoff for different widths of this layer.

## 3 METHOD

A standard feed-forward classification CNN can be formulated as

$$\mathcal{F} = \mathcal{C} \odot \mathcal{F}_{conv} = \mathcal{C} \odot (\mathcal{R}_L \odot f_L \odot \cdots \odot \mathcal{R}_1 \odot f_1), \tag{1}$$

where the overall function $\mathcal{F}$ is a composition of the convolutional feature extractor $\mathcal{F}_{conv}$ followed by a fully-connected classifier $\mathcal{C}$. The convolutional sub-model $\mathcal{F}_{conv}$ consists of a sequence of convolutional layers $[f_i : 1 \leq i \leq L]$, interspersed with non-linearities (ReLUs, Max-Pooling) or regularisers (dropout, BatchNorm) or some combination thereof, denoted by $\mathcal{R}_i$. The function performed by each convolutional layer $f_i$ is completely specified by a set of weights and biases that we denote using $W_i$. Crucially, the weights and biases for each different layer are independent. The number of parameters in layer $f_i$ is then simply the size of $W_i$, calculated as

$$|W_i| = n_i^{in} \times n_i^{out} \times k_i^2 + n_i^{out} = v_i \times k_i^2 + n_i^{out} \approx v_i \times k_i^2, \tag{2}$$

where $n_i^{in}$ in the number of input channels to $f_i$, $n_i^{out}$ is the number of output channels, $v_i = n_i^{in} \times n_i^{out}$ is the volume of $f_i$, and $k_i$ is the size of its (square) convolutional filters. In practice, the $n_i^{out}$ term for the biases is dominated by that for the weights, and so we disregard it in what follows. Letting $W_{conv} = \bigcup_{i=1}^{L} W_i$ denote all the parameters in $\mathcal{F}_{conv}$ (i.e. disregarding the comparatively small contributions from the non-convolutional layers), the total parameter count is given by

$$|W_{conv}| = \sum_{i=1}^{L} |W_i| \approx \sum_{i=1}^{L} v_i \times k_i^2. \tag{3}$$

Note that for many common architectures, there exists some $k$ such that $\forall i, k_i = k$ (e.g. for VGGNet, $k = 3$). For such architectures, Equation 3 can then be further simplified to $|W_{conv}| \approx L \times \bar{v} \times k^2$, in which $\bar{v} = L^{-1} \sum_{i=1}^{L} v_i$ is the mean volume per network layer.

Our method proposes a crude simplification to such architectures, namely to instantiate a single convolutional layer $f$, and apply it $L$ successive times in order to implement a convolutional pipeline of equivalent depth to the original model. In particular, we enforce the following constraint:

$$W_1 = W_2 = \cdots = W_L = W \Leftrightarrow f_1 = f_2 = \cdots = f_L = f. \tag{4}$$

This simplifies the CNN architecture in Equation 1 to

$$\tilde{\mathcal{F}} = \mathcal{C} \odot \tilde{\mathcal{F}}_{conv} = \mathcal{C} \odot (\mathcal{R}_L \odot f \odot \cdots \odot \mathcal{R}_1 \odot f). \tag{5}$$

Whilst our analysis focuses purely on the convolutional layers, it is interesting to note that when the $\mathcal{R}_i$ layers are all the same, the CNN architecture simplifies further to the following iterative form:

$$\tilde{\mathcal{F}} = \mathcal{C} \odot \tilde{\mathcal{F}}_{conv} = \mathcal{C} \odot (\mathcal{R} \odot f)^L. \tag{6}$$

The convolutional layer $f$ in our architecture expects an input tensor with a predetermined number of channels, which we will call $n$. Meanwhile, the $\mathcal{R}_i$ layers between the convolutional layers leave the number of channels unchanged. Thus, given the iterative application of $f$, the layer $f$ must also *output* a tensor with $n$ channels. (In practice, $f$ is called for the first time on the input image itself, which for colour images would normally only have 3 channels. To avoid artificially limiting $n$ to 3, we pad the input image with empty channels to produce a tensor with $n$ channels.) We deduce that $|W|$, the number of parameters for $f$, must satisfy $|W| \approx n^2 \times k^2 = v \times k^2$, where $v = n^2$ is the volume of $f$. Furthermore, since $W$ is shared between all $L$ convolutional layers, the total number of independent parameters in $\tilde{\mathcal{F}}_{conv}$ must also just be $|W|$. The compression factor between the original architecture and its shared counterpart can thus be quantified as

$$C = \frac{|W_{conv}|}{|W|} = \frac{\sum_{i=1}^{L} |W_i|}{|W|} \approx \frac{L \times \bar{v} \times k^2}{v \times k^2} = \frac{L}{v/\bar{v}}. \tag{7}$$

This is proportional to the depth $L$ of the original network, and is down-weighted by any (multiplicative) increase in the average per-layer volume in going from the original to the shared architecture.

We now turn to examine the convolutional operation in our architecture. Each layer $f$, the operation of which is completely specified by the weights and biases in $W$, takes an input tensor $X$ of size $n \times h \times w$, where $n$, $h$ and $w$ denote the number of channels, height and width respectively. Based on $X$ and $W$, we can conceptually define 2D matrices $\Phi(X)$ and $\Gamma(W)$ as follows:

$$\Phi(X) = \begin{bmatrix} \mathbf{x}_{11}^{\top} & \cdots & \mathbf{x}_{1n}^{\top} & 1 \\ . & & . & . \\ . & & . & . \\ \mathbf{x}_{m1}^{\top} & \cdots & \mathbf{x}_{mn}^{\top} & 1 \end{bmatrix}, \quad \Gamma(W) = \begin{bmatrix} \mathbf{w}_{11} & \mathbf{w}_{12} & \cdots & \mathbf{w}_{1n} \\ . & . & & . \\ . & . & & . \\ \mathbf{w}_{n1} & \mathbf{w}_{n2} & \cdots & \mathbf{w}_{nn} \\ b_1 & b_2 & \cdots & b_n \end{bmatrix}. \tag{8}$$

In this, $m = h \times w$, and each $\mathbf{x}_{ij}$ is a rasterisation of a $k \times k$ patch of input tensor centred at spatial location $i$ in channel $j$. Each $\mathbf{w}_{ij}$ is a similar rasterisation of the $k \times k$ convolutional kernel that maps the input channel $i \in \{1, 2, \ldots, n\}$ to the output channel $j \in \{1, 2, \ldots, n\}$, and each $b_j$ is the bias for output channel $j$. Then $f$ can be defined concisely as $f(X) = \Psi(\Phi(X) \times \Gamma(W))$, in which $\Psi$ reshapes the $m \times n$ tensor $\Phi(X) \times \Gamma(W)$ back to one of size $n \times h \times w$.

In practice, this simple formulation could be seen as being too restrictive, in the sense that irrespective of the convolutional iteration, each filter $\mathbf{w}_{ij}$ in $\Gamma(W)$ only ever operates on patches from input channel $i$ (for example, the $\mathbf{w}_{1j}$ filters only ever operate on patches from channel 1). For this reason, we decided to investigate whether adding a way of allowing the input channels to be reorganised at various points in the overall pipeline would improve performance. In principle, one way of achieving this would be to add $n \times n$ permutation matrices at appropriate points in the pipeline, e.g. just before each pooling operation. In practice, however, to make the operations differentiable, we implement them using linear layers (i.e. $1 \times 1$ convolutions), thus implementing blending of the input channels rather than simply permuting them. The weights of these layers are separate for each instantiation and are learned as part of the end-to-end pipeline.

It would be reasonable to expect this added flexibility to yield a significant increase in performance, and indeed our results in §5 show this to be the case. Nevertheless, it is notable that even without this added flexibility, our shared architectures already achieve extremely good performance on the datasets on which we tested, demonstrating that our underlying approach of sharing filters between layers makes sense even in the absence of permutation/blending.

## 4 DATASETS AND ARCHITECTURES

We evaluate our filter-sharing approach on four well-known image classification benchmarks: CIFAR-10, CIFAR-100, Tiny ImageNet and ImageNet. Details of these datasets can be found in §A.2. For this study, we work with two different architectures, one closely inspired by *VGGNet* (Simonyan & Zisserman, 2015), and the other by *ResNet* (He et al., 2016).

*VGGNet-like Architectures.* We base our VGGNet-like architectures on VGG-16, which consists of 5 convolutional blocks followed by 3 linear layers. Each block is followed by a max-pooling step and contains several convolutional layers with different channel counts (in order: 2 layers with 64 channels, 2 with 128, 3 with 256, 3 with 512 and 3 layers with 512 channels). By contrast, in our case, we define a single convolutional layer with a fixed number of input and output channels $n$, and then use it repeatedly in the same arrangement as above (see Table 3 in §A.3 for more details). We define four variants of this convolutional feature extractor for our study. *E-VGGNet* is our equivalent of VGGNet, with $n$ channels per layer and no sharing between the layers: we use this as a baseline. Its shared counterpart, *S-VGGNet*, has the same structure, but iteratively applies a single convolutional layer. *SL-VGGNet* is an extended version of *S-VGGNet* that introduces linear layers (i.e. $1 \times 1$ convolutions) before each max-pooling operation to allow the input channels to be blended at those points in the pipeline. Finally, since all the convolutional layers in *SL-VGGNet* are the same (these exclude what we call the linear layers), we define a further variant of our architecture that simplifies the network design by setting the number of layers per block to a scalar $\ell$. We experiment with $\ell \in \{2, 3\}$, and name the corresponding networks *SL$\ell$-VGGNet*. Note that the predetermined number of channels $n$ is a parameter of our architecture: we test several variants to find the best ones. We perform experiments on CIFAR-10/100 and Tiny ImageNet. For CIFAR-10, the 3 fully-connected layers that follow the feature extractor have 512, 512 and 10 output channels, respectively. For CIFAR-100, we use the same VGGNet-like architectures as for CIFAR-10, but the fully-connected layers have 1024, 1024 and 100 output channels, respectively. For Tiny ImageNet, we use a sequence of two fully-connected layers, with 2048 and 200 output channels respectively.

*ResNet-like Architectures.* We base our ResNet-like architectures on the models proposed in He et al. (2016). The simpler variants of these are built using 'basic' blocks that essentially consist of two equally-sized $3 \times 3$ convolutional layers and a skip connection (see Fig. 6). The deeper variants, meanwhile, are built using 'bottleneck' blocks, which similarly have a skip connection, but sandwich a single $3 \times 3$ convolutional layer between two $1 \times 1$ convolutional layers that decrease and then restore the number of channels to limit the number of free parameters. The network pipeline begins with a standalone convolutional layer that outputs a predetermined number of channels $p$. This is followed by a sequence of $b$ blocks at a number of different scale levels (generally 4, but 3 for CIFAR variants). In the original architectures, each scale level (except the first) began with a strided convolutional layer that downsampled the image and doubled the number of channels. Since we want the convolutional layers in our architectures to have the same numbers of input and output channels (to facilitate sharing), we define an equivalent architecture, *E-ResNet*, that instead doubles the number of channels and performs downsampling using (respectively) a linear layer (i.e. $1 \times 1$ convolutions) and a max-pooling step at the end of each scale level. Note that, as in the original ResNet, the final scale level in our architecture ends with average pooling rather than max-pooling. Despite these modifications, the predictive performances of our E-ResNets closely match those of the original architectures. The *shared* variant of this architecture uses $n$ channels for all scale levels and shares the weights across all the convolutional layers (excluding the linear layers). Since the architecture already contains the linear layers we were previously adding to allow blending of the input channels, we refer to it as *SL-ResNet*.

For CIFAR-10/100, the standalone convolutional layer uses a kernel size of $3 \times 3$, and a $p$ of 16 and 32 for each dataset, respectively. We experiment with $b \in \{3, 5, 7\}$ 'basic' blocks per scale level, and terminate the network with a 10-way linear classifier for CIFAR-10 and a 100-way classifier for CIFAR-100. See Table 4 in §A.3 for details. For Tiny ImageNet and ImageNet, we base our ResNet-like architectures on ResNet-34 and ResNet-50. ResNet-34 is built using 'basic' blocks, whilst ResNet-50 uses 'bottleneck' blocks. For the latter, it is clearly not possible to share filters between the layers within a block, since they are of different dimensions, so we instead use multiple shared copies of a single block. Note that the shared variants of both these models, SL-ResNet-34/50, keep the standalone convolutional layer unshared, since its kernel size is adjusted according to the dataset ($3 \times 3$ for Tiny ImageNet and $7 \times 7$ for ImageNet). See Table 5 in §A.3 for details.

## 5 RESULTS AND DISCUSSION

Earlier, Fig. 2 showed the accuracy vs. compression trade-off for S-VGGNet, relative to the original VGGNet (Simonyan & Zisserman, 2015), for different widths $n$ of the shared convolutional layer. Here, Fig. 3 illustrates the improvements in accuracy due to the learned linear layers (i.e. the blend-

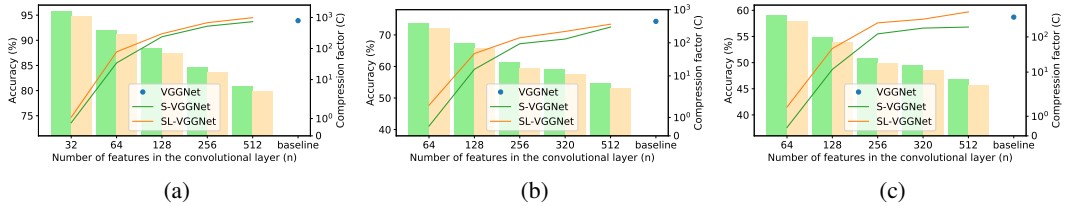

(a)    (b)    (c)

Figure 3: Accuracy versus compression trade-off curves of the 'S' and 'SL' variants of VGGNet for different widths $n$ of the shared convolutional layer, relative to the baseline VGGNet (Simonyan & Zisserman, 2015), for CIFAR-10 (a), CIFAR-100 (b), and Tiny ImageNet (c). The compression factor $C$ is plotted on a log scale.

| $n$ | E-VGGNet | | S-VGGNet | | | SL-VGGNet | | |
|---|---|---|---|---|---|---|---|---|
| | Acc. (%) | $\|W_{conv}\|$ | Acc. (%) | $\|W_{conv}\|$ | C | Acc. (%) | $\|W_{conv}\|$ | C |
| 32 | 87.2 | 112K | 73.6 | 9.3K | 12.0 | 74.7 | 13.3K | 8.4 |
| 64 | 91.2 | 445K | 85.5 | 37K | 12.0 | 87.7 | 53.3K | 8.3 |
| 128 | 93.0 | 1.8M | 90.7 | 148K | 12.2 | 91.3 | 213K | 8.4 |
| 256 | 94.1 | 7.1M | 92.8 | 590K | 12.0 | 93.5 | 852K | 8.3 |
| 512 | 94.5 | 28M | 93.7 | 2.4M | 11.7 | 94.5 | 3.4M | 8.2 |

(a) CIFAR-10: 'E' vs. 'S' vs. 'SL' variants

| Arch. | Acc. (%) | $\|W_{conv}\|$ | C |
|---|---|---|---|
| VGGNet* | 93.9 | 14.7M | 1.0 |
| Lego-VGGNet-16-w ($o$=2,$m$=0.5) | 93.23 | 3.7M | 4.0 |
| Lego-VGGNet-16-w ($o$=4,$m$=0.25) | 91.35 | 900K | 16.0 |
| SL2-VGGNet ($n$=512) | 94.5 | 3.4M | 4.3 |
| SL3-VGGNet ($n$=512) | 94.7 | 3.4M | 4.3 |
| SL2-VGGNet ($n$=256) | 93.7 | 852K | 17.3 |
| SL3-VGGNet ($n$=256) | 93.8 | 852K | 17.3 |

(b) CIFAR-10

| Arch. | Acc. (%) | $\|W_{conv}\|$ | C |
|---|---|---|---|
| VGGNet* | 74.3 | 14.7M | 1.0 |
| SL2-VGGNet ($n$=512) | 74.0 | 3.4M | 4.3 |
| SL3-VGGNet ($n$=512) | 74.4 | 3.4M | 4.3 |
| SL2-VGGNet ($n$=256) | 69.7 | 852K | 17.3 |
| SL3-VGGNet ($n$=256) | 71.0 | 852K | 17.3 |

(c) CIFAR-100

| Arch. | Top-1 (%) | Top-5 (%) | $\|W_{conv}\|$ | C |
|---|---|---|---|---|
| VGGNet* | 58.7 | 81.4 | 14.7M | 1.0 |
| SL2-VGGNet ($n$ = 512) | 59.4 | 82.8 | 3.4M | 4.3 |
| SL2-VGGNet ($n$ = 256) | 53.4 | 79.1 | 852K | 17.3 |

(d) Tiny ImageNet

Table 1: (a): Test accuracies and parameter counts $|W_{conv}|$ for our 'E', 'S' and 'SL' variants of *VGGNet*, for different widths $n$ of the convolutional layer. (b), (c), (d): Comparing the accuracies and compression factors $C$ of top-performing 'SL' variants of our approach, with $\ell \in 2, 3$ layers per convolutional block, with VGGNet (Simonyan & Zisserman, 2015) and (for CIFAR-10) variants of LegoNet (Yang et al., 2019b), another state-of-the-art compression method. Baseline models marked with a $*$ were retrained for this study.

ing layers) on CIFAR-10, CIFAR-100 and Tiny ImageNet. Observably, the use of the linear layers provides greater benefit for datasets that involve discriminating between a larger number of classes, such as CIFAR-100 and Tiny ImageNet.

For CIFAR-10, CIFAR-100 and Tiny ImageNet we compare the accuracies of the best-performing 'SL' variants of VGGNet with those of the baseline architecture (and competing compression methods for these datasets, where available) in Table 1. For CIFAR-10 (see Table 1b), we are able to achieve comparable classification accuracy to the VGGNet baseline using only $n = 256$ channels for our shared convolutional layer, which yields a compression factor of $\approx 17\times$. For CIFAR-100 (Table 1c), which has $10\times$ more classes, we had to use $n = 512$ channels to achieve comparable accuracy, but this still yields a significant compression factor of $4.3$. Higher compression factors can be achieved by reducing the number of channels, in exchange for some loss in accuracy. Evaluating our shared architecture on Tiny ImageNet (in Table 1d) evidences a similar trend in the results, with SL2-VGGNet ($n = 512$ channels) achieving an accuracy comparable to the non-shared baseline, whilst using only $23\%$ of its parameters. Detailed accuracy and memory usage numbers for E-VGGNet, S-VGGNet and SL-VGGNet, for CIFAR-10, are in Table 1a, while the results for CIFAR-100 and Tiny Imagenet can be found in the appendix (see Table 6 in §A.5)

We also evaluate our shared ResNet architecture (SL-ResNet) on Tiny ImageNet and ImageNet, with the results shown in Table 2 (the corresponding results for CIFAR-10 and CIFAR-100 can be found in the appendix, see Table 7 in §A.5). For Tiny ImageNet, our SL-ResNet34 ($n = 512$) variant is able to achieve a compression rate of $8.4$ with only a negligible loss in accuracy. For ImageNet, the same variant similarly achieves a compression rate of $8.4$ with respect to ResNet-50 and $21.6$ with respect to Shared Wide ResNet (SWRN) by Savarese & Maire (2019). Whilst there is

| Arch. | Top-1 (%) | Top-5 (%) | $|W_{conv}|$ | C |
|---|---|---|---|---|
| ResNet-50* | **62.8** | **84.1** | **26.8M** | **1.0** |
| ResNet-34* | 60.0 | 82.1 | 25.2M | 1.1 |
| SL-ResNet-50 ($n$=512) | 62.7 | 84.5 | 18.5M | 1.4 |
| SL-ResNet-50 ($n$=256) | 60.3 | 83.2 | 4.5M | 6.0 |
| SL-ResNet-34 ($n$=512) | **62.5** | **83.8** | **3.2M** | **8.4** |
| SL-ResNet-34 ($n$=256) | 56.2 | 80.0 | 794K | 33.6 |

(a) Tiny ImageNet

| Arch. | Top-1 (%) | Top-5 (%) | $|W_{conv}|$ | C |
|---|---|---|---|---|
| SWRN 50-2 | **78.26** | **94.05** | **69M** | **1.0** |
| ResNet-50 | 77.15 | 93.3 | 26.8M | 2.6 |
| ResNet-34 | 75.5 | 92.5 | 25.2M | 2.7 |
| ShaResNet-50 | 75.39 | 92.59 | 20.5M | 3.4 |
| ShaResNet-34 | 71.75 | 90.58 | 13.6M | 5.1 |
| Lego-Res50(o=2,m=0.5) | – | 89.7 | 8.1M | 8.5 |
| FSNet-ResNet-50 | 64.11 | 85.94 | 4.5M | 15.3 |
| SL-ResNet-50 ($n$=512) | 72.4 | 91.4 | 18.1M | 3.8 |
| SL-ResNet-34 ($n$=512) | **69.7** | **89.3** | **3.2M** | **21.6** |

(b) ImageNet

Table 2: Comparing the accuracies and compression factors $C$ of our shared variants of ResNet-34 and ResNet-50 with the original models and (for ImageNet) with ShaResNet (Boulch, 2018), LegoNet (Yang et al., 2019b), FSNet (Yang et al., 2019a) and Shared Wide ResNet (SWRN) (Savarese & Maire, 2019). Baseline models marked with a * were retrained for this study.

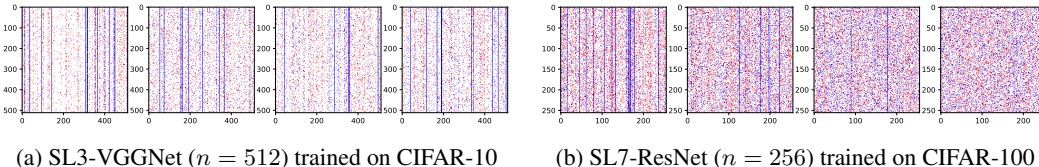

(a) SL3-VGGNet ($n = 512$) trained on CIFAR-10    (b) SL7-ResNet ($n = 256$) trained on CIFAR-100

Figure 4: A visual depiction of the linear layers used to blend the input channels in our approach. We show the layers for the two variants in the order (left to right) in which they appear in the networks. For each layer, the input channels are ordered along the x-axis, and the output channels along the y-axis. For each output channel (row), we highlight the lowest 32 weights (in terms of absolute value) in *blue*, and the highest 32 in *red*.

an accuracy trade-off, we achieve a greater compression rate than competing methods that achieve similar accuracies. Note that SWRN is able to achieve state-of-the-art levels of accuracy, but does not provide savings in the number of parameters.

## 5.1 INTERPRETATION THROUGH VISUALISATION

Visualising the weights of the blending layers that we learn for the SL-variants of our approach reveals interesting patterns in the way in which these layers blend (or use) the input channels (see Fig. 4). For each layer, the continuous *blue* vertical lines signify that a subset of the input feature maps are barely used by any of the output channels, thus effectively suppressing the information they carry. (Interestingly, the location of the vertical *blue* lines changes from one scale to the next, thus showing that different subsets of input channels go unused at different scales.) This is significant, because it implies that the weights associated with the unused channels can be selectively pruned without affecting performance. Our next experiment with the pruning method of Han et al. (2015) shows how we can exploit this observation to significantly reduce the size of our shared networks.

## 5.2 COMPLEMENTARITY WITH OTHER COMPRESSION SCHEMES

Our best-performing SL variants have a relatively small number of parameters in the convolutional layers, but a relatively high number of parameters in the linear layers. Tables 2a and 2b show how the parameter count for these variants increases with the number of channels $n$ and the depth (34 to 50). Notably, using bottleneck blocks, as we do for our SL-ResNet50 variants, also significantly increases the parameter count. As implied by our visualisations in the previous section, we would expect serious reductions in the number of parameters in the linear layers to be possible without significantly reducing accuracy. We thus experiment with applying the magnitude-based weight pruning approach of Han et al. (2015) to the linear layers to see whether this expectation is borne out in practice. We first select a proportion of the parameters to prune, then identify those weights that have the lowest absolute magnitude and set them to 0. We then evaluate on the validation split of the dataset. Note that we do not retrain the network after pruning. Our results (see Figure 5) show that we can remove a significant fraction of these blending weights before starting to see a noticeable drop in the accuracy of the network.

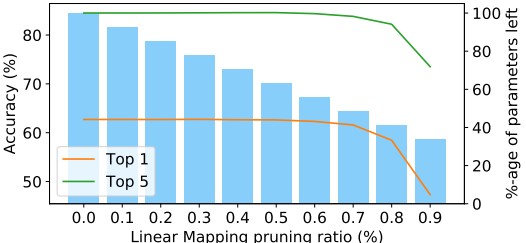

| Pruning Ratio (%) | Top 1 (%) | Top 5 (%) | $|W_{conv}|$ |
|---|---|---|---|
| 0 | 62.7 | 84.5 | 18.5M |
| 10 | 62.7 | 84.5 | 17.1M |
| 20 | 62.7 | 84.5 | 15.8M |
| 30 | 62.7 | 84.6 | 14.4M |
| 40 | 62.7 | 84.6 | 13.1M |
| 50 | 62.6 | 84.6 | 11.7M |
| 60 | 62.3 | 84.4 | 10.3M |
| 70 | 61.6 | 83.8 | 8.9M |
| 80 | 58.5 | 82.2 | 7.6M |
| 90 | 47.3 | 73.5 | 6.2M |

Figure 5: Analysing the effects of pruning on one of our largest models, SL-ResNet-50 ($n = 512$), trained on Tiny ImageNet. We iteratively zero out an increasing fraction of the linear layer parameters, starting from those having the smallest absolute value. The accuracy of the network stays constant even when $60\%$ of the parameters are pruned, at which point the compression rate (in comparison to the non-shared baseline with equivalent performance) has increased from $1.4$ to $2.6$.

## 6 CONCLUSION

In this paper, we leverage the regularities in visual signals across different scale levels to successfully extend the filter-sharing paradigm to an entire convolutional pipeline for feature extraction. In particular, we instantiate a single convolutional layer and apply it iteratively to simulate conventional *VGGNet-like* and *ResNet-like* architectures. We evaluate our shared architectures on four standard benchmarks – CIFAR-10, CIFAR-100, Tiny ImageNet and ImageNet – and achieve compression rates that are higher than existing sharing-based methods that have equivalent performance. We further show that even higher compression rates, with little additional loss in performance, can be achieved by combining our method with the magnitude-based weight pruning approach of Han et al. (2015). Study of our complementarity to more structured pruning techniques targeting complete filters and channels is reserved for future work. We conclude with two final observations. Firstly, our use of blending layers and a parameter to tune the width of the shared convolutional layer $n$ makes it easy to adjust the architecture so as to achieve a desired trade-off between compression rate $C$ and accuracy. Secondly, there are interesting connections between our work and the idea of energy-based pruning explored in (Yang et al., 2017), where the authors note that a significant fraction of the energy demands of deep network processing come from transferring weights to and from the file system. Our approach bypasses this bottleneck by using the same compact set of weights in an iterative manner. We aim to further investigate this aspect of our method in subsequent work.

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

# A    APPENDIX

## A.1    ADDITIONAL RELATED WORK

*Pruning* methods seek to reduce the size of a network by removing (either physically or implicitly) some of a network's weights (LeCun et al., 1990; Srinivas & Babu, 2015; Yang et al., 2017; Aghasi et al., 2017; Lee et al., 2019), filters (Leroux et al., 2016; Luo et al., 2017) or neurons (Ardakani et al., 2017). Notably, reducing the computational cost (rather than just the memory usage) of network architectures that are pruned in an unstructured manner requires the use of suitable sparse inference schemes.

*Quantization* methods keep the number of independent parameters in a network the same, but reduce the bit-depth of the parameters and activations (Wu et al., 2016; Hubara* et al., 2016; 2018) to limit the memory requirements of the network.

*Tensorization/tensor decomposition* methods propose low-rank approximations to high-dimensional neural matrices in order to downsize trained models. Early CNN architectures such as *AlexNet* (Krizhevsky et al., 2012) and *VGGNet* (Simonyan & Zisserman, 2015) contained the bulk of their weights in the fully-connected layers. As a result, various rank reduction approaches exclusively targeted the matrices in these layers (Lin et al., 2016; Novikov et al., 2015). The deeper/wider (He et al., 2016; Zagoruyko & Komodakis, 2016) these networks have become, the more the balance of weights has shifted towards the convolutional layers, giving rise to more generalised tensor decomposition schemes (Garipov et al., 2016).

*Knowledge distillation* ('teacher/student') methods aim to transfer the knowledge present in a cumbersome *teacher* model to a lightweight *student* model, without losing the teacher's ability to generalise well. An early approach by Bucilă et al. (2006) used a heavyweight ensemble to label a large set of unlabelled data, and then used this to train a compact model. Much later, Ba & Caruana (2014) proposed an alternative method that trains a shallow network to directly mimic the logits of a deep model. Subsequent methods have independently shown that training the student using temperature-scaled softmax scores (Hinton et al., 2014) or Gaussian-blurred logits (Sau & Balasubramanian, 2016) of the teacher can help with regularisation. Other methods in this line of work have proposed to train deep, thin neural networks using auxiliary or intermediate cues such as hidden layer outputs (Romero et al., 2015) or post-hoc attention maps (Zagoruyko & Komodakis, 2017).

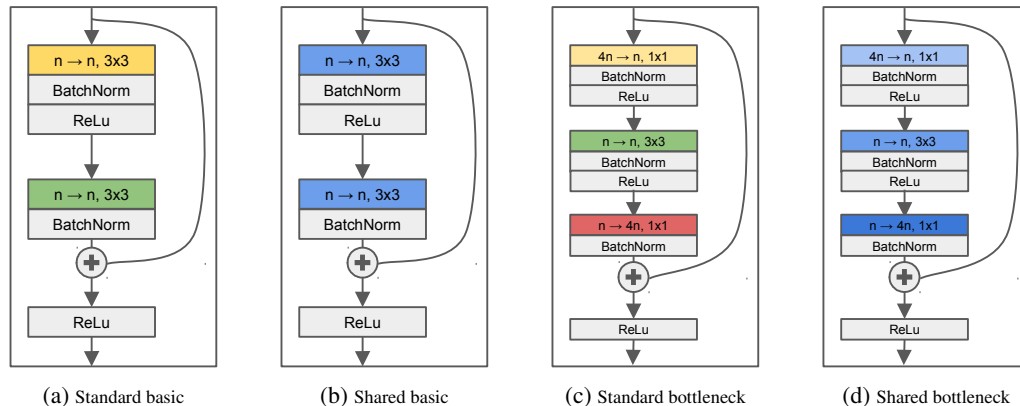

(a) Standard basic    (b) Shared basic    (c) Standard bottleneck    (d) Shared bottleneck

Figure 6: The building blocks used in our *ResNet-like* architectures. (a) and (b) show the non-shared and shared basic blocks. Our shared variant of the residual basic block reuses the same convolutional layer within and across the different blocks. (c) and (d) show the bottleneck blocks. In this case, since the three convolutions have different sizes, we cannot share a single set of parameters across the whole network; instead, we consider the block as a single entity and reuse it across the network.

*Custom architecture* methods, rather than trying to compress or distil knowledge from existing networks, propose entirely new network architectures that are smaller than existing models but still capable of providing excellent performance. Good examples include SqueezeNet (Iandola et al., 2016) and MobileNets (Howard et al., 2017). SqueezeNet tries to use $1 \times 1$ rather than $3 \times 3$ filters to reduce the parameter count, and tries to limit the number of input channels to those $3 \times 3$ filters it does use. MobileNets follow a similar tack and factorise traditional convolutional mappings into a depth-wise separable convolution (to process the spatial context) followed by a $1 \times 1$ convolution (to process the channels jointly). Two adjustable hyperparameters, $\alpha$ and $\rho$, pertaining to the intermediate feature resolution and the input spatial resolution, allow further resizing of the network.

*Hybrid* methods implement some combination of the compression schemes discussed above (Han et al., 2016; Yunpeng et al., 2017). Whilst our approach belongs to the category of filter-sharing schemes elaborated above, we also demonstrate its complementarity and compatibility with the magnitude-based weight pruning method of Han et al. (2015).

## A.2 DATASETS

CIFAR-10 (Krizhevsky, 2009) consists of $60,000$ $32 \times 32$ colour images, each labelled as belonging to one of 10 mutually exclusive classes. Each class contains $6,000$ images, of which $5,000$ are earmarked for training, and $1,000$ for testing (i.e. there are $50,000$ train images and $10,000$ test images overall). CIFAR-100 consists of the same $60,000$ $32 \times 32$ images that are in CIFAR-10, but this time they are evenly split into 100 classes, each containing 500 training images and 100 testing images. Tiny ImageNet[2] is essentially a smaller, lower-resolution variant of the ImageNet (Russakovsky et al., 2015) dataset. It consists of $120,000$ $64 \times 64$ images, evenly split into 200 classes. Each class contains 500 training images, 50 validation images and 50 test images. ImageNet (Russakovsky et al., 2015) was introduced as a large-scale image classification benchmark consisting of high-resolution photographs in $1,000$ visual categories from an even larger ontology of natural concepts (WordNet). It consists of approximately 1M training images, divided into $1,000$ disjoint object categories. Another set of $50,000$ images, evenly split into $1,000$ classes, forms the validation set. The accuracy results we report for ImageNet were obtained on this validation set.

## A.3 NETWORK ARCHITECTURES

Table 3 details the structure of our *VGGNet-like* architectures, whilst Tables 4 and 5 show our *ResNet-like* architectures (respectively used for CIFAR-10/100 and Tiny ImageNet/ImageNet). The notation is common to the tables and is as follows:

---

[2] https://tiny-imagenet.herokuapp.com

**conv1-**$x$  A $1 \times 1$ convolutional layer with $x$ output feature channels. The core of our SL variants. We use this layer to allow shared convolutions at different scale levels to observe different blends of the feature channels output by the previous scale. Its number of input feature channels is equal to $x$, except in E-ResNet-50, where we use it to increase the number of channels between scale levels, and in the first scale of SL-ResNet-50, where we use it to increase the number of channels from $x$ to $4x$, to account for the expansion factor of the bottleneck blocks.

**conv3-**$x$  A $3 \times 3$ convolutional layer with $x$ output feature channels. The number of input feature channels depends on the specific network variant: for the baselines it is equivalent to the number of output feature channels of the previous layer (or 3 for the very first layer), whilst for the E/S/SL-variants, it is equivalent to the number of output feature channels $x$. The stride is 1 unless otherwise specified.

**conv7-**$x$  A $7 \times 7$ convolutional layer with $x$ output feature channels. As this layer is only used as the first layer in the ResNet variants of our architectures, it always has 3 input channels. Its stride is 1 when training a *ResNet-like* architecture for Tiny ImageNet, and 2 when training for ImageNet.

**basicblock-**$x$  A simple skip connection-based block, used in the *ResNet-like* architectures. As in He et al. (2016), it consists of two $3 \times 3$ convolutional layers and a skip connection. In our shared architectures, the two convolutional layers share the same parameters. See Figures 6a and 6b for details of the internal architectures of the non-shared and shared block variants.

**bottleneck-**$x$  A skip connection-based block with a bottleneck architecture, consisting of a $1 \times 1$ convolution (used to reduce the number of feature channels), followed by a $3 \times 3$ convolutional layer, and finally by another $1 \times 1$ convolution (restoring the original number of feature channels). For this reason it has $4x$ input and output channels. Figures 6c and 6d detail the internal architectures of the standard and shared variants of the bottleneck blocks (respectively). Crucially, as mentioned in the main paper – and unlike the *basicblock* architectures described above – the bottleneck block is shared as a single entity, owing to the presence of differently-shaped convolutions.

**avgpool-**$x$  An average pooling layer operating on patches of size $x \times x$.

**maxpool-**$x$  A max-pooling layer operating on patches of size $x \times x$.

**FC-**$x$  A fully-connected layer with $x$ output channels. The number of its input channels is equal to the number of outputs of the previous layer (flattened in the case the previous layer was a convolutional layer).

Each spatial convolution (*conv3* and *conv7*) is always followed by a BatchNorm layer and a ReLu. We denote in **bold** the convolutional layers or blocks that are shared in our S and SL architectures. The parameters of the normalisation layers are never shared, even when the corresponding convolutional weights are shared as part of an S or SL architecture. Fully-connected layers (except the very last one in each architecture) are followed by a Dropout layer.

| Input Resolution | VGGNet | E-VGGNet($n$) | S-VGGNet($n$) | SL-VGGNet($n$) | SL$\ell$-VGGNet($n$) |
|---|---|---|---|---|---|
| CIFAR-*: $32 \times 32$
TI: $56 \times 56$ | conv3-64
conv3-64 | conv3-$n$
conv3-$n$ | **conv3-$n$**
**conv3-$n$** | **conv3-$n$**
**conv3-$n$**
conv1-$n$ | $\ell \times$**conv3-$n$**

conv1-$n$ |
| | | | maxpool-2 | | |
| CIFAR-*: $16 \times 16$
TI: $28 \times 28$ | conv3-128
conv3-128 | conv3-$n$
conv3-$n$ | **conv3-$n$**
**conv3-$n$** | **conv3-$n$**
**conv3-$n$**
conv1-$n$ | $\ell \times$**conv3-$n$**

conv1-$n$ |
| | | | maxpool-2 | | |
| CIFAR-*: $8 \times 8$
TI: $14 \times 14$ | conv3-256
conv3-256 | conv3-$n$
conv3-$n$ | **conv3-$n$**
**conv3-$n$** | **conv3-$n$**
**conv3-$n$**
conv1-$n$ | $\ell \times$**conv3-$n$**

conv1-$n$ |
| | | | maxpool-2 | | |
| CIFAR-*: $4 \times 4$
TI: $7 \times 7$ | conv3-512
conv3-512 | conv3-$n$
conv3-$n$ | **conv3-$n$**
**conv3-$n$** | **conv3-$n$**
**conv3-$n$**
conv1-$n$ | $\ell \times$**conv3-$n$**

conv1-$n$ |
| | | | maxpool-2 | | |
| CIFAR-*: $2 \times 2$
TI: $3 \times 3$ | conv3-512
conv3-512 | conv3-$n$
conv3-$n$ | **conv3-$n$**
**conv3-$n$** | **conv3-$n$**
**conv3-$n$** | $\ell \times$**conv3-$n$** |
| | | | maxpool-2* | | |
| | | | FC$_1$
FC$_2$
FC$_3$ | | |

Table 3: The architectures for *VGGNet* and the *VGGNet-like* networks we trained as part of our experiments on the CIFAR-10/100 and Tiny ImageNet datasets. The notation is described in the main text. Note that the last max-pooling layer (marked with a $*$) is not used when training a network for Tiny ImageNet: this is in order to provide a longer feature vector to the first fully-connected layer (specifically of size $n * 3 * 3$). The fully-connected layer sizes differ across datasets to account for the different numbers of classes, and are set as follows: (a) CIFAR-10: FC$_1$ = FC-512, FC$_2$ = FC-512, FC$_3$ = FC-10; (b) CIFAR-100: FC$_1$ = FC-1024, FC$_2$ = FC-1024, FC$_3$ = FC-100; (c) Tiny ImageNet: FC$_1$ = FC-2048, FC$_2$ = FC-200.

| Input Resolution | Eb-ResNet($p$) | SLb-ResNet($n$) |
|---|---|---|
| $32 \times 32$ | conv3-$p$
$b\times$ basicblock-$p$
conv1-$(2*p)$ | **conv3-$n$**
$b\times$ **basicblock-$n$**
conv1-$n$ |
| | maxpool-2 | |
| $16 \times 16$ | $b\times$ basicblock-$(2*p)$
conv1-$(4*p)$ | $b\times$ **basicblock-$n$**
conv1-$n$ |
| | maxpool-2 | |
| $8 \times 8$ | $b\times$ basicblock-$(4*p)$ | $b\times$ **basicblock-$n$** |
| | avgpool-8 | |
| | FC-$num_c$ | |

Table 4: The architectures for the *ResNet-like* networks we trained as part of our experiments on the CIFAR-10/100 datasets. The notation is described in the main text. The baselines E$b$-ResNet($p$) use $p = 16$ for training on CIFAR-10 (as in He et al. (2016)) and $p = 32$ for training on CIFAR-100. The final fully-connected layer has its output size set to the number of classes in the dataset (i.e. $num_c = 10$ for CIFAR-10 and $num_c = 100$ for CIFAR-100). We experiment with different values of $b \in \{3, 5, 7\}$.

| Input Resolution | E-ResNet-34 | E-ResNet-50 | SL-ResNet-34($n$) | SL-ResNet-50($n$) |
|---|---|---|---|---|
| $224 \times 224$ | conv7-64, stride-2 | conv7-64, stride-2
conv1-256 | conv7-$n$, stride 2 | conv7-$n$, stride 2
conv1-$(n*4)$ |
| $112 \times 112$ | maxpool-2 | maxpool-2 | maxpool-2 | maxpool-2 |
| $56 \times 56$ | $4\times$ basicblock-64
conv1-128 | $4\times$ bottleneck-64
conv1-512 | $4\times$ **basicblock-**$n$
conv1-$n$ | $4\times$ **bottneck-**$n$
conv1-$(n*4)$ |
| | maxpool-2 | maxpool-2 | maxpool-2 | maxpool-2 |
| $28 \times 28$ | $4\times$ basicblock-128
conv1-256 | $4\times$ bottleneck-128
conv1-1024 | $4\times$ **basicblock-**$n$
conv1-$n$ | $4\times$ **botteneck-**$n$
conv1-$(n*4)$ |
| | maxpool-2 | maxpool-2 | maxpool-2 | maxpool-2 |
| $14 \times 14$ | $4\times$ basicblock-256
conv1-512 | $4\times$ bottleneck-256
conv1-2048 | $4\times$ **basicblock-**$n$
conv1-$n$ | $4\times$ **botteneck-**$n$
conv1-$(n*4)$ |
| | maxpool-2 | maxpool-2 | maxpool-2 | maxpool-2 |
| $7 \times 7$ | $4\times$ basicblock-512 | $4\times$ bottleneck-512 | $4\times$ **basicblock-**$n$ | $4\times$ **botteneck-**$n$ |
| | avgpool-3 | avgpool-3 | avgpool-3 | avgpool-3 |
| | FC-$num_c$ | FC-$num_c$ | FC-$num_c$ | FC-$num_c$ |

Table 5: The architectures for the *ResNet-like* networks we trained as part of our experiments on the Tiny ImageNet and ImageNet datasets. The notation is described in the main text. The final fully-connected layer has its output size set to the number of classes in the dataset (i.e. $num_c = 200$ for Tiny ImageNet and $num_c = 1000$ for ImageNet). One important difference in the architectures for the two datasets is that, in the case of Tiny ImageNet, to account for the smaller resolution of the images, in the first scale level we use a $3 \times 3$ convolution without striding and suppress the first maxpool-2 layer. This has the effect of allowing us to feed the convolutional architecture with an input image of size $56 \times 56$.

| $n$ | E-VGGNet | | S-VGGNet | | | SL-VGGNet | | |
|---|---|---|---|---|---|---|---|---|
| | **Acc.** (%) | $|W_{conv}|$ | **Acc.** (%) | $|W_{conv}|$ | **C** | **Acc.** (%) | $|W_{conv}|$ | **C** |
| 64 | 64.4 | 445K | 41.1 | 37K | 12.0 | 47.7 | 53.3K | 8.3 |
| 128 | 70.8 | 1.8M | 59.1 | 148K | 12.2 | 64.1 | 213K | 8.4 |
| 256 | 74.6 | 7.1M | 67.2 | 590K | 12.0 | 69.1 | 852K | 8.3 |
| 320 | 75.3 | 11.1M | 68.7 | 922K | 12.0 | 71.1 | 1.3M | 8.5 |
| 512 | 76.9 | 28M | 72.5 | 2.4M | 11.7 | 73.4 | 3.4M | 8.2 |

(a) CIFAR-100

| $n$ | E-VGGNet | | S-VGGNet | | | SL-VGGNet | | |
|---|---|---|---|---|---|---|---|---|
| | **Acc.** (%) | $|W_{conv}|$ | **Acc.** (%) | $|W_{conv}|$ | **C** | **Acc.** (%) | $|W_{conv}|$ | **C** |
| 64 | 50.9 | 445K | 37.6 | 37K | 12.0 | 41.5 | 53.3K | 8.0 |
| 128 | 56.9 | 1.8M | 48.7 | 148K | 12.2 | 52.7 | 213K | 8.4 |
| 256 | 62.3 | 7.1M | 55.5 | 590K | 12.0 | 57.6 | 852K | 8.3 |
| 320 | 61.9 | 11.1M | 56.6 | 922K | 12.0 | 58.3 | 1.3M | 8.5 |
| 512 | 63.0 | 28M | 56.8 | 2.4M | 11.7 | 59.7 | 3.4M | 8.2 |

(b) Tiny ImageNet

Table 6: Test accuracies and parameter counts $|W_{conv}|$ for our 'E', 'S' and 'SL' variants of *VGGNet*, for different widths $n$ of the convolutional layer. The compression factors $C$ for the 'S' and 'SL' variants are computed relative to the corresponding *E-VGGNet*, which contains an equal number of channels $n$ in its convolutional layers. Note that all the models are trained from a state of random initialisation.

## A.4 Training Protocol

To train our networks on the CIFAR datasets, we perform some basic data augmentation steps: (1) we randomly decide whether or not to flip the input images horizontally, (2) we pad the $32 \times 32$ images with 4 pixels and then select a random crop of size $32 \times 32$, and finally (3) we normalise the RGB values to have zero mean and unit norm. During the evaluation phase, we just perform the normalisation step. We train our networks for 200 epochs, using the SGD optimiser with momentum 0.9 and weight decay $5e^{-4}$. We use an initial learning rate of 0.05 and decrease it by a factor of 2 when the error plateaus.

To train our networks on the Tiny ImageNet and ImageNet datasets, we perform a similar data augmentation: (1) we first extract a crop of a random size that is then resized to the input resolution of our network ($56 \times 56$ for Tiny ImageNet and $224 \times 224$ for ImageNet), (2) we randomly decide whether or not to perform a horizontal flip of the crop, and finally (3) we normalise the crop. During the evaluation phase, we (1) resize the image to a standard resolution ($64 \times 64$ for Tiny ImageNet and $256 \times 256$ for ImageNet), (2) extract ten crops (of size $56 \times 56$ for Tiny ImageNet and $224 \times 224$ for ImageNet) from the corners, the centre and their horizontally-mirrored variants (as in Krizhevsky et al. (2012)), and finally (3) normalise the crops. We train our networks for 100 epochs, using the SGD optimiser with momentum 0.9 and weight decay $5e^{-4}$. We use an initial learning rate of 0.01 for the *VGGNet-like* architectures on Tiny ImageNet, 0.05 for the *ResNet-like* architectures on Tiny ImageNet, and 0.1 for the experiments on ImageNet. Regardless of the initial value, we decrease it by a factor of 10 when the error plateaus.

## A.5 Additional Results

### A.5.1 Evaluation on Classification Benchmarks

Table 6 presents detailed accuracy and memory usage numbers for *E-VGGNet*, *S-VGGNet* and *SL-VGGNet* architectures trained on CIFAR-100 and Tiny ImageNet (results for CIFAR-10 can be found in the main paper, in Table 1a in §5). Similar results for the 'E' and 'SL' variants of ResNet trained on CIFAR-10 and CIFAR-100 can be found in Table 7. Finally, an accuracy and compression rate comparison of our top-performing *SL3-ResNet* variant with existing baselines and competing compression methods for CIFAR-10 is shown in Table 8.

| $b$ | E$b$-ResNet | | SL$b$-ResNet ($n=64$) | | SL$b$-ResNet ($n=96$) | | SL$b$-ResNet ($n=128$) | |
| --- | --- | --- | --- | --- | --- | --- | --- | --- |
| | **Acc.** (%) | $\|W_{conv}\|$ | **Acc.** (%) | $C$ | **Acc.** (%) | **C** | **Acc.** (%) | **C** |
| 3 | 91.8 | 294K | 89.7 | 6.5 | 92.2 | 2.9 | 93.1 | 1.6 |
| 5 | 92.9 | 488K | 90.1 | 10.8 | 92.0 | 4.8 | 93.0 | 2.7 |
| 7 | 93.4 | 682K | 89.6 | 15.2 | 91.9 | 6.7 | 93.2 | 3.8 |

(a) CIFAR-10

| $b$ | E$b$-ResNet | | SL$b$-ResNet ($n=128$) | | SL$b$-ResNet ($n=256$) | |
| --- | --- | --- | --- | --- | --- | --- |
| | **Acc.** (%) | $\|W_{conv}\|$ | **Acc.** (%) | **C** | **Acc.** (%) | **C** |
| 3 | 72.5 | 1.2M | 68.1 | 6.6 | 74.0 | 1.7 |
| 5 | 74.1 | 1.9M | 68.1 | 10.5 | 74.8 | 2.6 |
| 7 | 74.6 | 2.7M | 70.1 | 14.9 | 73.9 | 3.7 |

(b) CIFAR-100

Table 7: Test accuracies and parameter counts $|W_{conv}|$ for our 'E' and 'SL' variants of the ResNet architecture proposed for CIFAR-10 by He et al. (2016), for different widths $n$ of the convolutional layers and different number of blocks $b$ per scale level. The compression factors $C$ for the 'SL' variants are computed relative to their corresponding 'E' variants, which contain an equal number of blocks per scale level. Note that all the models are trained from a state of random initialisation.

| **Arch.** | **Acc.** (%) | $\|W_{conv}\|$ | **C** |
| --- | --- | --- | --- |
| ResNet-34 | 94.72 | 21.30M | 1.0 |
| ResNet-18 | 94.18 | 11.18M | 1.9 |
| ResNet* | 93.4 | 682K | 31.2 |
| FSNet-ResNet-18 | 93.93 | 810K | 26.3 |
| FSNet-ResNet-34 | 94.29 | 1.68M | 12.7 |
| FSNet-ResNet-50 | 94.91 | 2.51M | 8.5 |
| ShaResNet-164 (Boulch, 2018) | 93.8 | 0.93M | 23.0 |
| SL3-ResNet ($n=128$) | 93.1 | 181K | 117.7 |

Table 8: CIFAR-10: Comparing the accuracies and compression factors $C$ of top-performing 'SL' variant of the ResNet architecture (He et al., 2016), for $b=3$ blocks per scale level, with the original ResNet, other baselines ResNet-18 and ResNet-34, and state-of-the-art compression methods. The compression factor of the proposed model with respect to the best performing ResNet-34 architecture is in triple digits. However, a more appropriate comparison is arguably with ResNet*, from which the model has been directly compressed by virtue of sharing the convolutional layers. The compression factor is still a significant 4.0, with a final weight count of only 181K. Note that the model marked with a ∗ has been retrained for this study.

A.5.2 INTERPRETATION THROUGH VISUALISATION

In Fig. 7, we show the linear layers for our different variants of *VGGNet*, trained on three different datasets – CIFAR-10, CIFAR-100 and Tiny ImageNet. As highlighted by the continuous *blue* vertical lines, it is notable that in each layer, some of the input channels barely contribute towards any of the output channels. Given this, we posit that a significant proportion of the weights in the linear layers (those that apply to the least important input channels) can be pruned without affecting the accuracy in any significant manner. Preliminary results, verifying this conjecture, are discussed in §5.1. Interestingly, the changing locations of these *blue* lines reflects the changing importance of different input channels at different scale levels.

Similar results for four different 'SL' variants of ResNet, trained on three different datasets – CIFAR-10, CIFAR-100 and Tiny ImageNet – are presented in Fig. 8. As with our visualisations for 'SL-VGGNet', the continuous *blue* vertical lines in Figs. 8b, 8c and 8d highlight that some input channels make only a minimal contribution to any of the output channels in each layer. Once again, we believe that the weights that are applied to these less-important input channels can be pruned without affecting the accuracy in any significant manner. Some indicative results that support this hypothesis can be found in §5.1. By contrast, the linear layers in Fig. 8a exhibit somewhat less regularity. From Table 7a, SL7-ResNet yields both the highest accuracy (93.2%), and the highest compression rate (3.8) for that accuracy amongst all the variants. Thus, one possible explanation for this regular distribution of linear layer weights is that the model is operating at full capacity and is using all the channels in a balanced way to achieve an optimal performance.

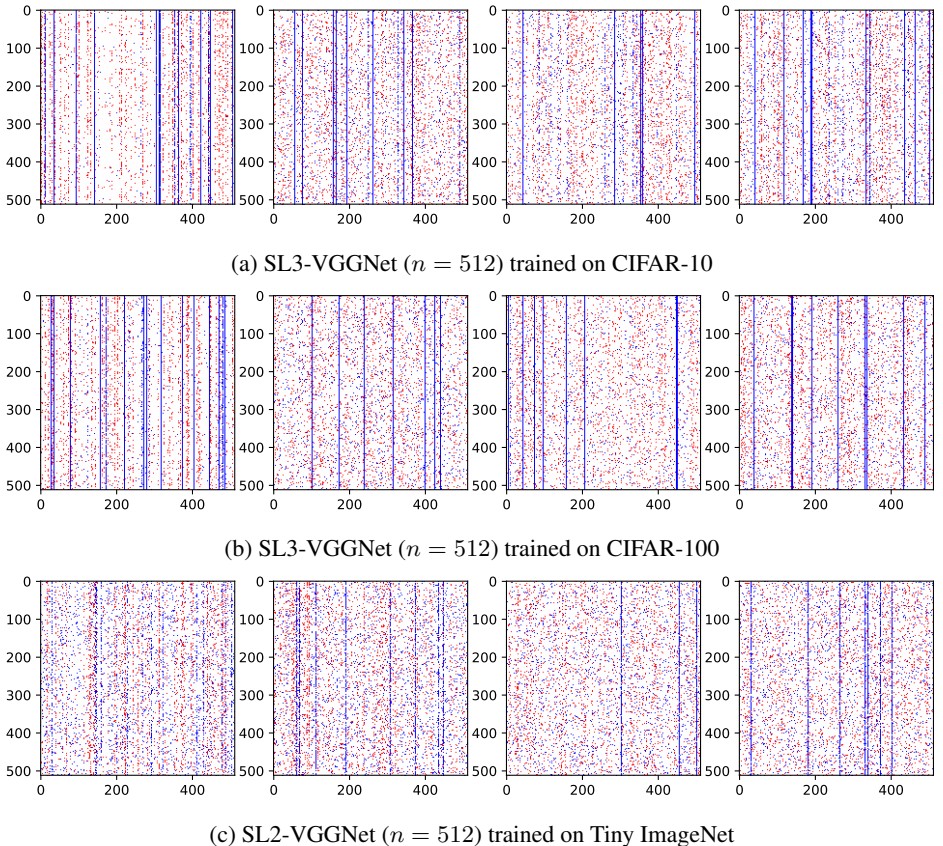

(a) SL3-VGGNet ($n = 512$) trained on CIFAR-10

(b) SL3-VGGNet ($n = 512$) trained on CIFAR-100

(c) SL2-VGGNet ($n = 512$) trained on Tiny ImageNet

Figure 7: A visual depiction of the linear layers used to blend the input channels in the 'SL' variants of *VGGNet* trained on CIFAR-10, CIFAR-100 and Tiny ImageNet. The linear layers are presented in the order (left to right) in which they appear in the networks. For each layer, the input channels are ordered along the x-axis, and the output channels along the y-axis. For each output channel (row), we highlight the lowest 32 weights (in terms of absolute value) in *blue*, and the highest 32 in *red*.

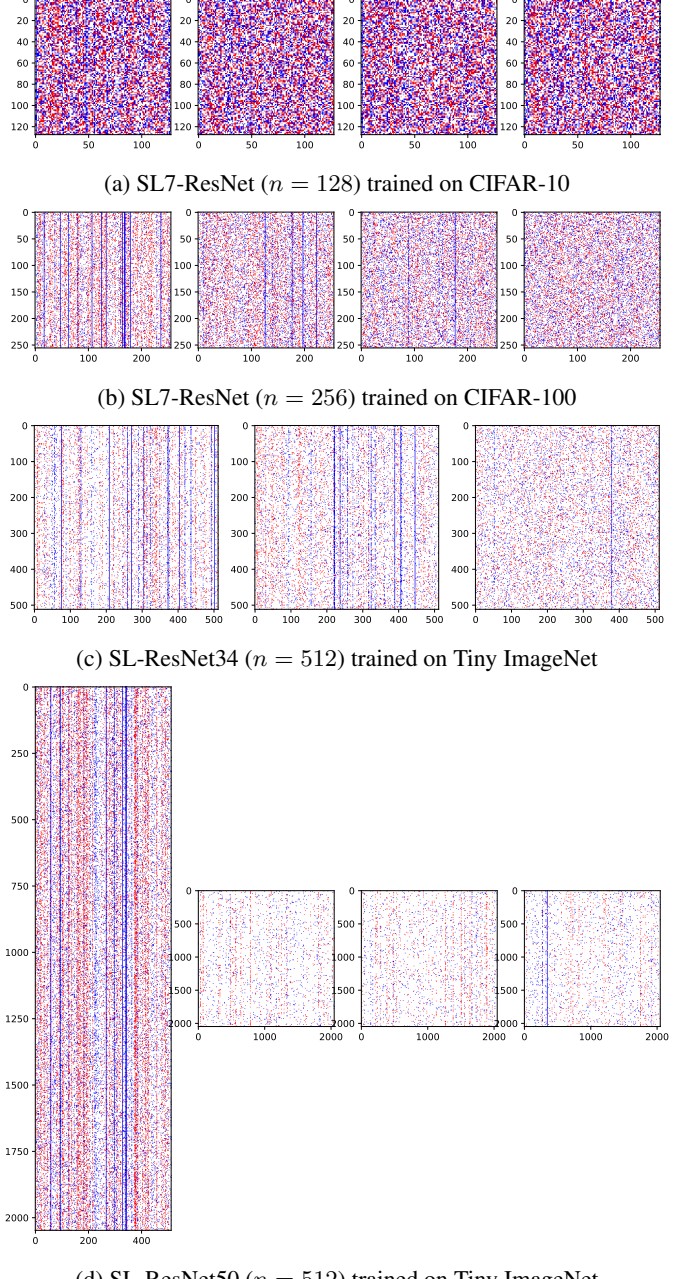

(a) SL7-ResNet ($n = 128$) trained on CIFAR-10

(b) SL7-ResNet ($n = 256$) trained on CIFAR-100

(c) SL-ResNet34 ($n = 512$) trained on Tiny ImageNet

(d) SL-ResNet50 ($n = 512$) trained on Tiny ImageNet

Figure 8: A visual depiction of the linear layers used to blend the input channels in four 'SL' variants of *ResNet*, in the order (left to right) in which they appear in the networks. For each layer, the input channels are ordered along the x-axis, and the output channels along the y-axis. For each output channel (row), we highlight the lowest 32 weights (in terms of absolute value) in *blue*, and the highest 32 in *red*.

