# OpenReview forum: "ShardNet: One Filter Set to Rule Them All"
_ICLR.cc/2020/Conference — Reject_

### Official Review · AnonReviewer2 · 2019-10-21
**Official Blind Review #2**

**Rating:** 3

**Review:**

In this paper, the authors propose to use the *same* convolutional layer in every layer of a DNN. The network effectively is converted into repeatedly applying the same convolutional filter at multiple scales. The idea is motivated by wavelet decompositions and related work. The authors show that by repeatedly applying the same filter, the number of parameters that need to be stored for a model reduces proportionally to the depth of the network. At the same time, experimental evidence is provided that the performance of these models is not affected, when compared to the baseline (full) model.

COMMENTS:
- the paper is well written, but overly verbose. there are several areas where the explanation can be compressed, making room to add more informative details (which are in the appendix), or increasing the size of the figures (which are too small)

- the paper seems lacking a bit in experiments. If the authors can show that the same filter applied L times achieves about the same performance, why not also experiment with different L? i.e. does VGGNet actually need L layers? what if only 2 layers are used? this will help with the overparametrization problem as well.

- Figures 3 and 4 are hard to read. Please increase their size.

- page 5 line 2: how does padding the input with (n-3) empty channels affect performance? If you're learning the filters through backprop, will they not always be learning to fit to 0 ? or am i missing something?

- along the above lines, why not have the input layer to be a different filter with 3 channels and then have a common filter for all upstream layers?

- in multiple places in the text, you refer to the number of "independent" parameters. I dont see why the parameters need to be independent. Unless there's some orthogonalization happening at the weights, calling them independent is incorrect.

- paragraph above sec4: you add separate "linear" layers for the 'SL' models. Can you describe how many addditional parameters you have to learn?

**Experience Assessment:**

I do not know much about this area.

**Review Assessment: Checking Correctness Of Derivations And Theory:**

I carefully checked the derivations and theory.

**Review Assessment: Checking Correctness Of Experiments:**

I carefully checked the experiments.

**Review Assessment: Thoroughness In Paper Reading:**

I read the paper at least twice and used my best judgement in assessing the paper.

---

> ### Author Response · Authors · 2019-11-13
> **Response to Reviewer 2**
>
> We thank the reviewer for their useful feedback. We have identified some parts in the introduction and related work that could be shortened in the interest of making space for some interesting results which are at present in the appendix. This would also allow us to increase the size of Figures 3 and 4.
>
> “why not also experiment with different L. does VGGNet actually need L layers? what if only 2 layers are used? this will help with the overparametrization problem as well.”
>
> Simply decreasing the number of layers L would require us to increase the size of the filters by a significant amount if we are to draw the class inference from the entire image and not a small patch of the input image. Thus, any naive reduction in the number of layers would not help solve the problem of over-parameterisation, but might aggravate it instead. Historically, in moving from AlexNet to VGGNet, it has been observed that deeper networks with smaller filter sizes offer a good tradeoff between network footprint and performance. ith ResNets[], the filter sizes were made smaller still, and the pipeline was made much deeper while using skip-connections to alleviate the problem of vanishing gradients. This resulted in compact and highly effective network architectures.
> Further, several recent studies [1, 2] have attempted to understand when and why ‘deeper networks are more effective than shallower ones’. Our work doesn’t aim to answer this question. Instead, we present a unique approach to compressing deep networks without compromising the benefits of depth. We aim to achieve similar performance to a complex, multi-layered pipeline by iteratively applying a single layer (which can be seen as a shallow function). We demonstrate the efficacy of this idea for both plain feed-forward constructs and residual constructs. That said, it should be noted that some of the experiments we did perform did involve the use of networks of different depths. In particular, we show results for VGGNets with different depths in Tables 1 and 6, and for ResNets with different depths in Tables 2 and 8.
>
> “If you're learning the filters through backprop, will they not always be learning to fit to 0 ?”
>
> To simplify our experimental setup, we keep the number of filters at the first layer the same as at other layers. This amounts to applying some filters over inputs channels that are anchored to 0. Note once again that the filter set is shared across all layers and receives different inputs at different layers of the network. Since the input to the filters in question may not be 0 at other layers, these filters that are redundant at layer 0 can still capture information that is relevant for class disentanglement at layers that are further down the network hierarchy.
>
> “why not have the input layer to be a different filter with 3 channels and then have a common filter for all upstream layers?”
>
> Using a non-shared set of filter weights for the first layer did not make much difference for the small datasets. For Tiny ImageNet and ImageNet, this indeed became relevant on two accounts - (a) to add flexibility through untied weights (b) to regulate the spatial resolution differently at the beginning of the pipeline and for the rest of the pipeline. Thus, our experimental setup was adapted to incorporate a standalone set of weights for the first layer. Please see the last paragraph of Section 4 on page 6 that states the following - "Note that the shared variants of both these models, SL-ResNet34/50, keep the standalone convolutional layer unshared, since its kernel size is adjusted according to the dataset (3 × 3 for Tiny ImageNet and 7 × 7 for ImageNet)."
>
>
> “- in multiple places in the text, you refer to the number of "independent" parameters. I dont see why the parameters need to be independent.”
>
> Some papers have referred to the parameters as independent, for instance [3]. This may be because, under proper regularisation, the number of non-zero weights can be thought to define the dimensionality of the function space of the network i.e. the number of axes of the function space. But the authors agree that the use of the word independent is a slight abuse of terminology. We will replace this with the word ‘individual’.
>
> “you add separate "linear" layers for the 'SL' models. Can you describe how many additional parameters you have to learn?”
>
> The count of additional parameters introduced by the linear layers for VGGNet variants can  be calculated as the difference between the number of parameters for the SL and S variants in Table 6 in the appendix. We will update the caption accordingly to make this clear.
>
> [1] Learning Functions: When Is Deep Better Than Shallow, Hrushikesh Mhaskar, Qianli Liao, Tomaso Poggio
> [2] On the Number of Linear Regions of Deep Neural Networks, Guido Montúfar, Razvan Pascanu, Kyunghyun Cho, Yoshua Bengio
> [3] Learning Implicitly Recurrent CNNs Through Parameter Sharing Pedro Savarese, Michael Maire ICLR 2019

---

### Official Review · AnonReviewer1 · 2019-10-22
**Official Blind Review #1**

**Rating:** 3

**Review:**

This paper presents an approach to reduce the number of a neural network by sharing the convolutional weights among layers. To convert the first layer into the right number of features padding is used. The last layer I suppose is instead a normal classifier on the fully connected representation (for VGG) or on the average pooling(for ResNet). Results on different datasets and architectures show that the proposed approach can highly compress the number of needed parameters with a minimal reduction of the network test accuracy.

I lean to reject this paper because, in my opinion is very similar to ("Bridging the Gaps Between Residual Learning, Recurrent Neural Networks and Visual Cortex" Qianli Liao and Tomaso Poggio), which is not mentioned in related work. This paper, published in 2016 was already proposing the idea of reducing the number of parameters of ResNet by sharing the weights of each layer and therefore consider ResNet with shared weights as a recurrent net.
In this paper the setting are slightly different, authors add also a variant with additional 1x1 convolutions and show also results with additional compression. However, in my opinion, the main idea is the sharing of convolutional weights, and this is not new.


Additional Comments:
- This paper considers only the number of learnable parameters of a network. However, in many cases, for applications, it is more important to save memory (which is not the case as the activations should still be saved for backpropagation) and computation. In my understanding the final computation of the model is actually increased because it uses more channels at lower layers (which corresponds to high resolution features maps). Authors should comment about that.
- In section 4, VGGNet-like Architectures and ResNet-like architectures the authors mention a baseline E-VGGNet or E-ResNet with exactly the same architecture as the shared weights network (thus same number of channels at each layer), but without sharing. However I could not find the performance of that interesting baseline in the results.






**Experience Assessment:**

I have published in this field for several years.

**Review Assessment: Checking Correctness Of Derivations And Theory:**

I assessed the sensibility of the derivations and theory.

**Review Assessment: Checking Correctness Of Experiments:**

I assessed the sensibility of the experiments.

**Review Assessment: Thoroughness In Paper Reading:**

I read the paper at least twice and used my best judgement in assessing the paper.

---

> ### Author Response · Authors · 2019-11-11
> **Response to Reviewer 1**
>
> We thank the reviewer for their helpful feedback and for the reference to the Qianli Liao and Tomaso Poggio paper. Indeed this work is relevant to our current submission and we are working towards discussing it in the paper. Please find a detailed response contextualising our work and highlighting its novelty vis-a-vis the reference in Reply #1. Further, it is true that the idea of sharing convolutional weights is not new, we mention several such methods in the related work, but we believe that our approach is sufficiently different from those other methods to qualify as novel. The rest of the points concern clarifications, and we are working towards answering them in our next post.

---

> ### Author Response · Authors · 2019-11-13
> **Response to Reviewer 1: additional comments**
>
> "This paper considers only the number of learnable parameters of a network. However, in many cases, for applications, it is more important to save memory (which is not the case as the activations should still be saved for backpropagation) and computation. In my understanding the final computation of the model is actually increased because it uses more channels at lower layers (which corresponds to high resolution features maps). Authors should comment about that."
>
> We consider the memory and computational requirements of our approach at both training and inference time, in comparison to the baseline networks. Note that for real-world applications, it is common to focus more on the costs at inference time, since networks are commonly trained on powerful multi-GPU clusters, and so an increase in memory and/or compute requirements at training time usually does not present insuperable problems.
> At training time, our requirements for both are higher, owing to our reliance on a single convolutional filter having a higher number of channels. That requires more memory and computation, due to the fact that backpropagated gradients have to be computed and stored for all the layers and weights.
> At inference time, the activations and gradients no longer have to be stored, which already significantly reduces the memory requirements. Additional memory can be saved by simply loading a single copy of the shared layer (which can then be applied repeatedly), in which case our overall memory usage will be much less than the baseline (which is not able to do this). This is useful in memory-constrained environments, such as are common in deployment scenarios (e.g. self-driving cars, robotics, smartphones, etc.). Here, the baseline would have to repeatedly swap individual layers in and out of memory, incurring significant I/O overhead. However, our approach would not suffer from this problem. This gives our approach a significant advantage in such scenarios.
>
> "In section 4, VGGNet-like Architectures and ResNet-like architectures the authors mention a baseline E-VGGNet or E-ResNet with exactly the same architecture as the shared weights network (thus same number of channels at each layer), but without sharing. However I could not find the performance of that interesting baseline in the results."
>
> The relevant results can be found in Tables 6 and 8 in section A.4.1 of the appendix. In the paper as it stands, these are mentioned at the end of Section 5. However, we accept that the way in which they were mentioned was slightly too general, and we will update the text to reference the tables explicitly in order to make the results more discoverable.

---

### Official Review · AnonReviewer3 · 2019-10-23
**Official Blind Review #3**

**Rating:** 1

**Review:**

This paper proposes to modify a standard CNN by requiring all of its layers to share the same filter set, essentially allowing it to be expressed as an iterative (or recurrent) network.  This also has the effect of forcing the same number of feature channels to be used throughout the network.  For ResNet-like architectures with bottleneck blocks, sharing occurs at the level of the block (3 conv layers in series that are repeated).  Another variant of the sharing pattern inserts unshared 1x1 convolutional layers after shared layers or blocks; this adds some flexibility while still reducing parameters compared to standard CNNs.

On CIFAR-10, CIFAR-100, and Tiny ImageNet, experiments demonstrate the ability of the sharing scheme to reduce parameters without impacting accuracy (or more drastically reduce parameters at the cost of accuracy) (Tables 1ab, 2a).

However, results are less compelling on ImageNet (Table 2b), where SL-ResNet-50 and SL-ResNet-34 are both less accurate than the baseline standard ResNets as well as ShaResNet [Boulch, 2018].  The accuracy gap between SL-ResNet and ResNet on ImageNet (Table 2b) is significant (approx 5% Top-1 and 2% Top-5 accuracy) and might make it difficult to justify use of the proposed method in this setting.  As ImageNet is the most challenging of the datasets used, this is cause for concern.

There is also a major concern with respect to novelty and related work.  Unfortunately, the paper appears to have completely missed the following highly related publication from ICLR 2019:

Learning Implicitly Recurrent CNNs Through Parameter Sharing
Pedro Savarese, Michael Maire
ICLR 2019

This prior work proposes a network structure in which a set of L layers share a set of k parameter templates.  The templates and sharing coefficients are learned as part of the standard training procedure.  This prior work demonstrates both parameter savings and accuracy improvements when training networks in this manner.  Additionally, this prior work shows that some learned networks can be converted into explicitly recurrent forms as a post-processing step.

The paper under review appears be a special case of this prior work with the number of templates k = 1 (shared between all layers).  It is possible this is an important special case, worthy of significant attention on its own.  Notably, [Savarese and Maire, 2019] considered sharing across at most all layers within the same stage of a residual network, rather than all layers in the network.  However, arguing for the importance of this special case would require focused experimental comparison and analysis, which is not present in the current version of the paper.

Novelty is clearly limited in light of this overlooked prior work.  At minimum, citation, discussion, and experimental comparison to the above ICLR 2019 paper is necessary.

**Experience Assessment:**

I have published in this field for several years.

**Review Assessment: Checking Correctness Of Derivations And Theory:**

N/A

**Review Assessment: Checking Correctness Of Experiments:**

I carefully checked the experiments.

**Review Assessment: Thoroughness In Paper Reading:**

I read the paper thoroughly.

---

> ### Author Response · Authors · 2019-11-11
> **Response to Reviewer 3**
>
> We thank the reviewer for their helpful feedback and for the reference to the Savarese and Maire paper which we unfortunately missed. We are now in the process of integrating a comparison to it in our submission. For the reviewer’s concerns regarding novelty please see our Reply #1.

---

### Author Response · Authors · 2019-11-11
**Reply #1: Related Work and Novelty 1/2**

We thank reviewers R1 and R3 for drawing our attention to [1] and [2] respectively. These studies are indeed highly relevant to our current work and we regret not having seen them earlier. We are now in the process of adding a comparison to their methodology and findings in our submission. In the meantime, we would like to emphasise that the existence of this literature does not diminish the novelty of our work but in fact adds to it. We elaborate further as follows.
The authors of [1] discuss ResNets that share weights between individual residual units as implementing a time-invariant homogeneous function and thus being a generalisation of RNNs. On the experimental front, however, they only ever analyse CIFAR-10. When experimenting with weight-sharing across time, the transition functions for different states are implemented using non-shared convolutional layers and the recurrence function comprises of a predetermined number of self-transitions interspersed with transitions across states. In this regime, their 2-state fully recurrent network achieves an accuracy vs. size tradeoff of ~90% for 298K parameters. In the further case when they examine weight-sharing across states (using a single set of convolutional weights across all layers for a 3-state shared ResNet using 64 features), they achieve an accuracy of ~85% for 40K parameters. This weight-sharing scheme closely matches that of our Shared-VGGNet variant with 64 features. Our model is able to achieve 85.5% accuracy for 37K parameters (See Table 6(a) in Appendix), and we are able to obtain comparable results without ever using any residual connections. Further, the formulation of [1] either treats the transition functions (convolutional layers) as being either the same or different, there is never any soft-association between the different transition functions. In comparison, for our SL variant, we can consider the combination of a linear layer followed by a convolutional layer as a composite layer. These composite layers (which are equivalent to the transition functions in [1]) are then associated with the shared layers by virtue of their 2D filter maps being a linear combination of the shared filter maps. This association lends greater flexibility to our model without incurring huge cost in parameter count. Observably, the SL variant for VGGNet with 64 features achieves an accuracy of 87.7% for 53.3K parameters (See Table 6(a) in Appendix), and the SL variant of ResNet with equal number features achieves an accuracy of 89.7% for 45.3K parameters (See Table 8(a) in Appendix) - a significant improvement in performance over the basic model of [1] without a huge increase in the number of parameters.
The above described formulation (of the SL variant) is also what differentiates our work from [2]. The authors of [2] propose to model each convolutional layer as a linear combination of similarly-sized layer templates from a bank (they use one bank per size or scale level). As observed by the reviewer, if all the layers in their network had the same size, and if they had used precisely one bank containing a single template (a case that was not addressed in the original work), then their formulation would have mapped to our S-variant. However, in any case, we take this paradigm further with the SL variant by modeling the 2D filter maps of some layers as a linear combination of the shared filter maps. Notice also that in terms of the experimental work, [2] only ever analyses Wide-ResNet architecture. For CIFAR dataset, their model achieves 96% for 12M parameters. Compare this to our ~94% accuracy for 0.8M parameters. Clearly, the approach of [2] doesn’t focus on network compression while our work aims to achieve a competitive balance between accuracy and parameter count. Particularly for ImageNet, a most challenging vision dataset as noted by the reviewers, their model doesn’t demonstrate any parameter saving whilst showing a marginal 0.26% (top-1) and 0.1% (top-5)  increase in accuracy for a total of 69M parameters. In comparison, we investigate the compression aspects of our approach for ImageNet and present two shared variants of ResNet with 3.8 and 23 times fewer parameters than [2] (18.1M and 3.2M in total, respectively) while losing a few points in accuracy; it is this tradeoff that is a crucial contribution of our study (as further evidenced by the results we report in Table 2b, where we show that our accuracy is comparable to that of other “sharing-based” approaches, while achieving a higher compression rate).

---

> ### Author Response · Authors · 2019-11-11
> **Reply #1: Related Work and Novelty 2/2**
>
> In conclusion, when viewed in the light of these two additional studies our work can be seen as even more timely. While there is some overlap between the S variant of our approach and a special case of [2], our SL variant goes beyond it and provides additional flexibility. For [1], we would like to quote a sentence from their discussion - “A radical conjecture would be: the effectiveness of most of the deep feedforward neural networks, including but not limited to ResNet, can be attributed to their ability to approximate recurrent computations that are prevalent in most tasks with larger than shallow feedforward networks. This may offer a new perspective on the theoretical pursuit of the long-standing question “why is deep better than shallow””. Our study puts this conjecture to the test of experimental rigour for two state-of-the art deep network architectures - VGGNet and ResNet - and a variety of visual datasets including CIFAR, TinyImageNet and ImageNet, while opening new pathways of thinking about the simplifications in conventional deep nets and how they might be reconciled with biological models.
>
> [1]    Bridging the Gaps Between Residual Learning, Recurrent Neural Networks and Visual Cortex Qianli Liao and Tomaso Poggio
> [2]    Learning Implicitly Recurrent CNNs Through Parameter Sharing Pedro Savarese, Michael Maire ICLR 2019

---

### Author Response · Authors · 2019-11-15
**Revised Paper**

We thank the reviewers again for their useful feedback. As promised, we have now uploaded a revised version of the paper that contains:
- A more concise version of the introduction.
- A revision to the ‘Related Work’ section to include discussion about the latest literature on recurrent implementations of convolutional neural networks.
- An update to the ‘Results’ section to make the results of our E variants more discoverable.
- Increased the size of figures to make them more readable.

We believe that the updated version of the manuscript makes it clearer how our work compares to existing recurrent CNN approaches, and better highlights the novel contributions made by our approach.

---

### Decision · Program_Chairs · 2019-12-19

**Decision:**

Reject

**Comment:**

This submission proposes an interesting experiment/modification of CNNs. However, it looks like this contribution overlaps significantly with prior work (that the authors initially missed) and the comparison in the (revised) manuscript seem to not clearly delineate and acknowledge the similarities and differences.

I suggest the authors improve this aspect and try submitting this work to next venue.